# Hospital costs of Balloon Pulmonary Angioplasty (BPA) procedure and management for CTEPH patients: An observational study based on the French national hospital discharge database (PMSI)

**Vincent Cottin[1]***, **Lionel Bensimon[2]**, **Fanny Raguideau[3]**, **Gwendoline Chaize[3]**,
**Antoinette Hakmé[2]**, **Laurie Levy-Bachelot[2]**, **Alexandre Vainchtock[3]**, **Jean Dallongeville[4]**,
**Hélène Bouvaist[5]**, **Philippe Brenot[6]**

1 National Coordinating Reference Center for Rare Pulmonary Diseases, Louis Pradel Hospital, University of Lyon, INRAE, Lyon, France, 2 MSD France, Puteaux, France, 3 HEVA, Lyon, France, 4 Pasteur Institute, INSERM U1167, Lille, France, 5 Grenoble University Hospital, Grenoble, France, 6 Marie Lannelongue Hospital, Le Plessis-Robinson, France

* vincent.cottin@chu-lyon.fr

**Data Availability Statement:** A regards data availability, due to the sensitive nature of the data

## Abstract

### Introduction

Since 2014, Balloon Pulmonary Angioplasty (BPA) has become an emerging and complementary strategy for chronic thromboembolic hypertension (CTEPH) patients who are not suitable for pulmonary endarterectomy (PEA) or who have recurrent symptoms after the PEA procedure.

### Objective

To assess the hospital cost of BPA sessions and management in CTEPH patients.

### Methods

An observational retrospective cohort study of CTEPH-adults hospitalized for a BPA between January 1st, 2014 and June 30th, 2016 was conducted in the 2 centres performing BPA in France (Paris Sud and Grenoble) using the French national hospital discharge database (PMSI-MCO). Patients were followed until 6 months or death, whichever occurred first. Follow-up stays were classified as stays with BPA sessions, for BPA management or for CTEPH management based on a pre-defined algorithm and a medical review using type of diagnosis (ICD-10), delay from last BPA procedure stay and length of stay. Hospital costs (including medical transports) were estimated from National Health Insurance perspective using published official French tariffs from 2014 to 2016 and expressed in 2017 Euros.

### Results

A total of 191 patients were analysed; mainly male (53%), with a mean age of 64,3 years. The first BPA session was performed 1.1 years in median (IQR 0.3–2.92) after the first PH

that support the findings, access to them is restricted and can only be granted by the Ethics and scientific committee for health research, studies, and evaluations (CESREES, Comité Ethique et Scientifique pour les Recherches, les Etudes et les Evaluations dans le domaine de la Santé) and/or the French data protection authority (Comité National de l'Informatique et des Libertés, CNIL), and so are not readily available. The data are part of the National health data system (SNDS, Système national des Données de Santé) a database maintained by the HDH (Health Data Hub). The HDH can be contacted at https://www.health-data-hub.fr/contact.

**Funding:** This study was financed by MSD. The funder provided support in the form of salaries for authors [LB,AH and LLB], but did not have any additional role in the study design, data collection and analysis, decision to publish, or preparation of the manuscript. LB, AH and LLB are employees of MSD and participated to the study design, study analysis, decision to publish and preparation of the manuscript. AV is one of the co-founders of the CRO HEVA; FR and GC are employees of HEVA, a company who received funding from the study sponsor for the conduct of this study. VC, JD, HB and PB are independent experts who received fees for participating in the scientific committee of the study.

**Competing interests:** I have read the journal's policy and the authors of this manuscript have the following competing interests: Experts: VC, JD, HB and PB are independent experts who received fees for participating in the scientific committee of the study. Outside the submitted work, VC reports personal fees and non-financial support from Actelion, grants, personal fees and non-financial support from Boehringer Ingelheim, personal fees from Bayer / MSD, personal fees from Novartis, personal fees and non-financial support from Roche / Promedior, personal fees from Sanofi, 2 personal fees from Celgene / BMS, personal fees from Galapagos, personal fees from Galecto, personal fees from Shionogi, personal fees from Astra Zeneca, personal fees from Fibrogen, personal fees from RedX, personal fees from PureTech. Outside the submitted work, HB has received personal fees and/or non-financial support from MSD, Actelion, GSK. Outside the submitted work, PB and JD have no competing interest in relation with this topic to declare. MSD employees: LB, AH and LLB are employees of MSD, the company which financed the study. HEVA employees: AV is one of the co-founders of the CRO HEVA; FR and GC are employees of HEVA, a company who received funding from the study

hospitalisation. A mean of 3 stays with BPA sessions per patient were reported with a mean length of stay of 8 days for the first stay and 6 days for successive stays. The total hospital cost attributable to BPA was € 4,057,825 corresponding to €8,764±3,435 per stay and €21,245±12,843 per patient. Results were sensitive to age classes, density of commune of residence and some comorbidities.

## Conclusions

The study generated robust real-world data to assess the hospital cost of BPA sessions and management in CTEPH patients within its first years of implementation in France.

## Introduction

Pulmonary hypertension (PH) is a disease defined by the increase of mean pulmonary arterial pressure ≥20 mm Hg [1] Although this definition is the same across PH, subtypes of different prognosis and therapeutic implications exist due to differences in underlying mechanisms [2]. Chronic thromboembolic pulmonary hypertension (CTEPH) is a rare subtype of PH (Group 4), characterized by the presence of organized fibrous tissue (chronic thromboembolic material) occluding varying degrees of the pulmonary arterial tree which may completely block the lumen [3]. In France, the prevalence has been recently estimated at 47 cases per million [4]. The patient's survival at 1 year, 2 years, and 3 years are approximately 93%, 91%, and 89%, respectively [5].

Management of CTEPH in France is organized around a national reference centre, and affiliated regional centres in university hospitals (known as competence centres). Patients should be evaluated by a multidisciplinary team for suitability for pulmonary endarterectomy (PEA) which may be curative. However, about 40–50% are ineligible to receive this surgical intervention due to distal lesion, advanced age or comorbidities, and some patients may still experience symptoms after the procedure. For these patients, medical treatment of CTEPH and/or Balloon Pulmonary Angioplasty (BPA) are recommended [2, 6]. Furthermore, Riociguat, an oral guanylate cyclase stimulator, and treprostinil, a subcutaneous prostacyclin analogue, are approved (in 2014 and 2020 respectively) for patients with inoperable CTEPH or persistent/recurrent PH after PEA; other PH medications are also off-label used [7]. BPA has been considered recently as an emerging and complementary strategy for CTEPH patients who are not suitable for PEA or CTEPH patients who have recurrent symptoms after the PEA procedure [8, 9]. BPA is an interventional procedure consisting of repeat sessions of catheterizations and dilatations using a balloon catheter to progressively enlarge pulmonary stenosis and recover optimal hemodynamic and pulmonary perfusion [8]. This intervention is less invasive than PEA [10]. The technique of BPA has been reported in 2001 and was improved over the years with the technical advances and the accumulating experience in the management of pulmonary artery stenosis [8]. Given those results, the European Society of Cardiology/European Respiratory Society 2015 recommended BPA *"for patients who are technically inoperable or who carry an unfavourable risk/benefit ratio for PEA (class IIb recommendation, level of evidence C)"* [2, 3]. Recommendations specified that BPA should only be performed in experienced and high volume CTEPH centres [2]. In France, the BPA procedure has been initiated since 2014 in two centres, Paris Sud University (Bicetre Hospital and Marie Lannelongue Hospital) and Grenoble University Hospital.

sponsor for the conduct of this study and for the
analysis of the data. These commercial affiliations
does not alter adherence to all PLOS ONE policies
on sharing data and materials.

Despite the efficacy of BPA and evidence of improvement on short-term symptoms, exercise capacity and hemodynamics [11], there is no data to describe the costs attributable to BPA. Procedure and management are in a learning curve, but it is not clear what their costs are due to recurrence of sessions and risk of complications. Thus, understanding BPA-related costs seems of key importance. To do so, the present study aimed to generate real-world data to assess the hospital cost of BPA sessions and management in patients with CTEPH within its first years of implementation. This study was conducted from the French national health insurance perspective.

## Materials and methods

### Study design

This was an observational retrospective cohort study of -adult patients with CTEPH hospitalized for a BPA procedure between January 1st, 2014 and June 30th, 2016 (inclusion period, i.e. first years of implementation). As BPA was introduced in France in 2014, all included patients were incident, i.e. included at time of their first BPA session occurring in the 2 French centres performing BPA during the inclusion period (index date). BPA protocol is the same in both centers. Patient spent around 6 days at the hospital with an overnight admission for two BPA sessions repeated within 48 hours.

Patients were followed up during a period of 6 months that we considered relevant to assess cost attributable to BPA session and management—or death, whichever occurred first.

### Data sources

The French PMSI-MCO (Medicine, Surgery, Obstetrics) database covers all overnight or day hospitalisations in the public and private sectors involving short-term stays in medical, surgical or obstetric facilities. Each patient in the database is attributed a unique anonymous patient identifier. This identifier can be used to track individual patients across multiple hospitalisations.

At the time of final discharge, a standardised discharge summary (SDS) is issued which lists all hospital procedures undergone by the patient during the stay, identified through standardised procedure codes. The reason for hospitalisation is identified by a diagnosis-related group (DRG) code, based on the International Classification of Diseases, 10th revision (ICD-10) [12], which is used by the hospital administration for costing purposes. Three different types of DRG code may be attributed to an individual stay. The principal diagnosis (PD) corresponds to the condition for which the patient was hospitalised (for example, myocardial infarction); the related diagnosis (RD) corresponds to any underlying condition which may have been related to the PD (for example, coronary artery disease); the significantly-associated diagnosis (SAD) corresponds to comorbidities or complications which may affect the course or cost of hospitalisation (for example, chronic kidney failure). All medical procedures are listed, including surgery, diagnostic tests and other examinations, and the departments in which the patient was hospitalised during the stay are documented. Most medications and non-pharmacological treatments cannot be specifically identified since they are integrated into the DRG cost. However, delivery of certain expensive drugs and recorded in a linked database (FICH-COMP) and can thus be identified individually. Socio-demographic information is limited to gender, age and postcode of residence. No information is available on the outcome of any procedure or the results of any test.

## Study population

We included all adult patients ($\geq$18 years) with at least one hospitalisation for PH (ICD-10 codes I270 and I272) as principal, related or significantly associated diagnosis and with an associated BPA procedure (CCAM codes DFAF001, DFAF002, DFAF003, DFAF004) between January 1st, 2014 and June 30th, 2016 in the 2 centres performing BPA in France (Paris Sud and Grenoble). In the absence of specific ICD-10 codes for CTEPH, we assumed that our combination of selection criteria included only patients with CTEPH, due to the exclusive indication of BPA in this PH subtype. Patients with a PEA and a BPA procedure during the same hospital stay were however excluded, considering in this context BPA as a rescue procedure directly linked to the PEA instead of the CTEPH management. We excluded also patients having BPA stays with DRG codes for lung transplant or level 4 cardiothoracic surgical intervention, as costs of these stays are mainly driven by other costs than BPA (outliers stays). Comorbidities were identified based on a medical review of ICD-10 codes reported in hospital stays occurring within the year prior to inclusion (ICD-10 codes are reported in S1 File).

## Healthcare resource utilization and costs

Stays were classified into three categories: for BPA sessions, for management of BPA complications or for CTEPH management. Stays with BPA sessions (inclusion and additional stays) were identified as stays with both a diagnosis code (as PD or RD or SAD) for PH and a BPA procedure performed in Paris Sud and/or Grenoble, as per inclusion criteria. Other stays were categorized based on expert's medical review using the combination of the following criteria: type of diagnosis (ICD-10 codes), delay from last BPA session and length of stay. In case of difficulty to differentiate stays for BPA management or for CTEPH management based on ICD-10 codes and length of stay, a 30-days delay from the last BPA session was considered: hospital stays occurring within 30 days after BPA sessions were categorized as stays related to BPA management; hospital stays occurring more than 30 days after BPA sessions were categorized as stays related to CTEPH management and not directly related to BPA management.

Costing was restricted to direct costs and determined from the perspective of the French social security system (National Health Insurance; NHI). All costs were expressed in Euros. Costs were attributed from official French national tariffs for medical acts applicable in France from 2014 to 2016. These costs were updated to 2017 values to take inflation into account. A standard national tariff was applied to each hospitalization based on the DRG code attributed in the PMSI database. These standard tariffs include medical and related procedures, nursing care, treatments (except specific expensive drugs), food and accommodation, and investment costs for hospitalised patients. Additional costs per day of hospitalisation in an intensive care unit were added to the DRG tariffs when appropriate [13]. For private hospitals, where physicians are reimbursed on a fee-for-service basis, physician fees were identified from the ENCC (*Echelle Nationale des Coûts à Méthodologie Commune*, the French observatory of real-world spending on healthcare) and added to the DRG tariffs. Expensive drugs were costed using the public retail price. Out of hospital drugs were not included in the present analysis as they are not recorded in the database. The cost of medical transports was also estimated using transports fees for each mode of transport (ambulance, light medical vehicle, taxi, public transport, and personal vehicle) in each department and considering the distance between commune of residence and hospital. As mode of transport is not available in PMSI, the transport cost was then weighted according to the repartition of transport mode published by NHI in 2016 for patients $\geq$ 60 years with long term diseases (Open DAMIR). The full methodology is detailed in S2 File.

## Statistical analyses

Descriptive analyses were performed. Continuous data were presented as mean values with standard deviation (SD) or median values with range (quartile 1 and quartile 3) and categorical data as frequency counts and percentages. Comparative statistical analyses were performed to compare characteristics of patients according to year of inclusion (based on index date of each patient). The Chi-2 test (significant level, 5%) was used for qualitative variables; the Student's t-test or the Wilcoxon test depending on the nature of the data distribution was used for quantitative variables (significant level, 5%). Random-effect regression models were performed to explore predictive factors associated with costs, considering a factor with random-effect (hospital centre). Statistical Analysis System software, version 9.2 for Windows (SAS Institute Inc., Cary, NC, USA) was used for all analyses.

## Ethics

The study was conducted in accordance with International Society for Pharmacoepidemiology (ISPE) Guidelines for good pharmacoepidemiology practices (GPP) and applicable regulatory requirements. Because this was a retrospective study using an anonymized database and had no influence on patient care, ethics committee approval was not required. Use of the PMSI-MCO database for this type of study has been approved by the French national data protection agency (CNIL; annual authorisation #1419102 v8–2015-111111-56-18 / order M14L056).

# Results

Between January 1st, 2014 and June 30th, 2016, 198 CTEPH adult patients with BPA sessions were identified in the PMSI MCO-database (Fig 1). We excluded 4 patients having a BPA and a PEA procedure performed during the same stay and 3 patients with outliers stays, as per protocol. Finally, 191 CTEPH patients were included (n = 129 in Paris Sud and n = 62 in Grenoble). The inclusion increased over years: 45 patients were included in 2014, 92 in 2015 and 54 in mid-2016 (up to 30/06/2016).

## Patients' characteristics

At baseline, the mean age was 64.3 years (SD = 14.7) and 53.4% were male (Table 1). A history of PEA procedure was found for 3.1% of patients. The time interval between the first hospitalization for PH (since 2006) and the first stay for a BPA procedure was a mean of 2.1 years (SD = 2.5) and a median of 1.1 years (IQR 0.3–2.9). Among comorbidities, chronic respiratory disease was reported for 31.4% (n = 60) of patients, heart failure for 28.3% (n = 54), a hospitalization with obesity diagnosis for 14.7% (n = 28), a diabetes mellitus for 9.9% (n = 19), a chronic ischemic heart disease for 9.4% (n = 18) and a chronic renal disease for 6.3% (n = 12). Characteristics of patients were not significantly different according to year of inclusion. The proportion of patients according to quartiles of social deprivation index was respectively, from the most deprived to the most privileged, 16.7%, 23.6%, 25.3% and 34.4%. Half of patients (50.5%) lived in urban environment, 23.4% in semi urban environment, 18.6% in semi-rural environment and 7.5% in rural environment.

## Healthcare resource utilization

Over the 6 month-follow-up period, patients had several additional stays either for BPA sessions, BPA management or CTEPH management. A total of 146 patients (76.4%) had at least one additional stay with BPA sessions (270 stays, i.e. 1.8 stays per patient), 2 patients (1.0%)

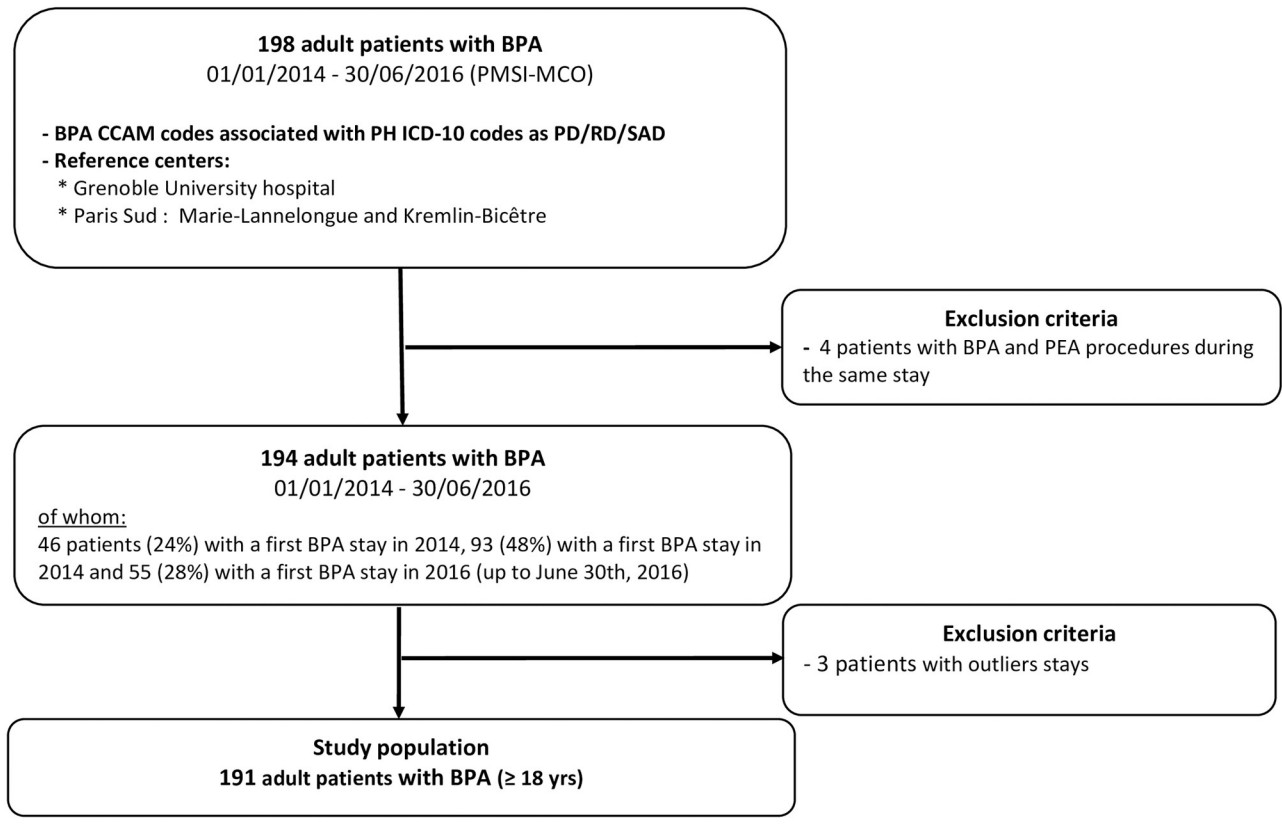

**Fig 1. Flow chart of the study population.**

had one additional stay for BPA management (2 stays, 1 stay per patient) and 81 patients had at least one additional stay for CTEPH management (103 stays, i.e 1,3 stays per patient) (Table 2). Thus, patients had an average of 2.8 stays with BPA sessions over the study period (including inclusion BPA session). All stays with BPA sessions were overnight hospitalizations and 88.5% of them had an Intensive care unit (ICU) transit. CTEPH management stays were also mainly (84.5%) overnight hospitalizations.

The first stays with BPA sessions (inclusion stays) had a mean length of 7.8 days (SD = 4.5) including a mean length of stay in ICU of 4.8 days (SD = 3.6) whereas follow-up stays with BPA sessions had a mean length of 6.0 days (SD = 2.3). Moreover, length of BPA stays decreased over the years with a mean of 8.5 days (SD = 4.6) in 2014, 7.9 (SD = 3.8) in 2015 and 6.9 (SD = 5.4) in 2016. Follow-up stays for BPA management and CTEPH management were shorter with a mean length of 2.0 days (SD = 1.4) and 3.7 days (SD = 3.7), respectively.

## Cost

Over the study period, the total cost of hospital care for CTEPH adult patients experienced BPA was € 4,343,886. This cost included mainly (€4,057,825, 93%) hospital care related to BPA sessions or management: €1,781,596 for inclusion stays (€ 335,442 in ICU; 18,8%), €2,271,707 for follow-up stays (€255,428 in ICU, 11,2%) and €4,522 for BPA management (€ 322 in ICU;7.1%) (Table 2). CTEPH management accounted for 7% of the total cost, €286,061.

The mean BPA related cost per stay was €8,764 (SD = 3,435): BPA sessions had a mean cost per stay of €9,328 (SD = €3,989) at inclusion (including €1,756 (SD = € 1,521) for ICU), €8,414

**Table 1. Demographic and medical characteristics of patients at baseline.**

|  | Total = 191 |
| --- | --- |
| Age, years, mean (SD) | 64.3 (14.7) |
| Male, n (%) | 102 (53.4%) |
| Social Deprivation index, n (%) |  |
| Most deprived | 31 (16.7%) |
| Deprived | 44 (23.6%) |
| Privileged | 47 (25.3%) |
| Most privileged | 64 (34.4%) |
| Missing | 5 |
| Population density index, n (%) |  |
| Rural | 14 (7.5%) |
| Semi rural | 35 (18.6%) |
| Semi urban | 44 (23.4%) |
| Urban | 95 (50.5%) |
| Missing | 3 |
| History of PEA*, n (%) | 6 (3.1%) |
| PH duration*, years, mean (SD) | 2.1 (2.5) |
| Median (IQR) | 1.1 (0.3–2.9) |
| At least one comorbidity among*,**, n (%) |  |
| Ascites | 3 (1.6%) |
| Chronic ischaemic heart disease | 18 (9.4%) |
| Chronic renal disease | 12 (6.3%) |
| Chronic respiratory disease | 60 (31.4%) |
| Coagulation disorder | 7 (3.7%) |
| Connective tissue disease | 6 (3.1%) |
| Diabetes mellitus | 19 (9.9%) |
| Heart failure | 54 (28.3%) |
| Hypothyroidism | 9 (4.7%) |
| Liver disease | 5 (2.6%) |
| Obesity | 28 (14.7%) |
| Pacemaker | 4 (2.1%) |
| Sickle cell disease | 3 (1.6%) |

*Information available since 2006

** ICD-10 codes used to track comorbidities are reported in S1 File.

(SD = €2,890) during follow-up (including €946 (SD = € 968) for ICU) and BPA management had a mean cost per stay of €2,261 (SD = €1,702) (including € 161 (SD = € 228) for ICU). CTEPH management had a mean cost per stay of €2,777 (SD = €2,437). These costs tended to decrease over the years (Fig 2).

The mean BPA related cost per patient was €21,245 (SD = €12,843) including €2,101 (SD = €1,948) of transportation cost: BPA sessions had a mean cost per patient of €9,328 (SD = €3,989) at inclusion, €15,560 (SD = 10,108) during follow-up and BPA management has a mean cost per patient of €2,261 (SD = 1,702). CTEPH management had a mean cost per patient of €3,532 (SD = €3,891).

Results were also sensitive to age classes, density of commune of residence and some comorbidities. Age intervals of 46–55 years and 56–65 years were significantly associated with a high cost of BPA compared to an age of 18–25 years, after adjusting for sex, density of

**Table 2. Healthcare resource use and cost of hospital stays.**

| | BPA-related | | | CTEPH management related |
|---|---|---|---|---|
| | Sessions at inclusion | Sessions during follow-up | Complications | |
| Number of patients, n | 191 (100%) | 146 (76.4%) | 2 (1,0) | 81 |
| Number of stays per patient, n | 1.0 | 1.8 | 1.0 | |
| Day hospitalizations, n (%) | 0 (0.0%) | 0 (0.0%) | 0 (0.0%) | 16 (15.5%) |
| Overnight hospitalizations, n (%) | 191 (100.0%) | 270 (100.0%) | 2 (100.0%) | 87 (84.5%) |
| Length of overnight hospitalization, in days, mean (SD) | 7.8 (4.5) | 6.0 (2.3) | 2.0 (1.4) | 3.7 (3.7) |
| Number of stays with ICU, n (%) | 169 (88.5%) | 162 (60.0%) | 1 (50.0%) | 7 (6.8%) |
| Total cost* | | € 4,057,825 | | € 286,061 |
| | € 1,781,596 | € 2,271,707 | € 4,522 | |
| Part of ICU among total cost (%) | € 335,442 (18.8%) | € 255,428 (11.2%) | € 322 (7.1%) | € 15,016 (5.2%) |
| Cost/stay, mean (SD)* | | € 8,764 (€ 3,435) | | € 2,777 (€ 2,437) |
| | € 9,328 (€ 3,989) | € 8,414 (€ 2,890) | € 2,261 (€ 1,702) | |
| ICU cost/stay, mean (SD) | € 1,756 (€ 1,521) | € 946 (€ 968) | € 161 (€ 228) | € 146 (€ 860) |
| Cost /patient, mean (SD)* | | € 21,245 (€ 12,843) | | € 3,532 (€ 3,891) |
| | € 9,328 (€ 3,989) | € 15,560 (€10,108) | € 2,261 (€ 1,702) | |
| ICU cost /patient, mean (SD) | € 1,756 (€ 1,521) | € 1,750 (€ 1,679) | € 161 (€ 228) | € 185 (€ 967) |

*For total cost, cost per stay and cost per patient, the first row provides a pooled costs of stays at inclusion, during follow-up and for management of complications; the second row provides a costs per type of stays, respectively.

commune, liver disease and chronic respiratory disease, with respectively an adjusted RR of 1.3 [95% CI:1.0–1.8] and 1.7 [95% CI:1.3–2.3] (Fig 3). No significance was reached for other age classes. Low, high and very high density of commune of residence was associated with a low cost of BPA, compared to very low density of commune of residence with respectively adjusted RR of 0.7 [95% CI:0.5–0.9], 0.8 [95% CI:0.6–1.1] and 0.6 [95% CI:0.5–0.9]. Finally, liver diseases were significantly associated with a high cost of BPA with an adjusted RR 1.8 [95% CI:1.1–2.9]. Chronic respiratory diseases tended to be associated with a low cost of BPA but without significance (RR = 0.8 (95% CI:0.7–1.0).

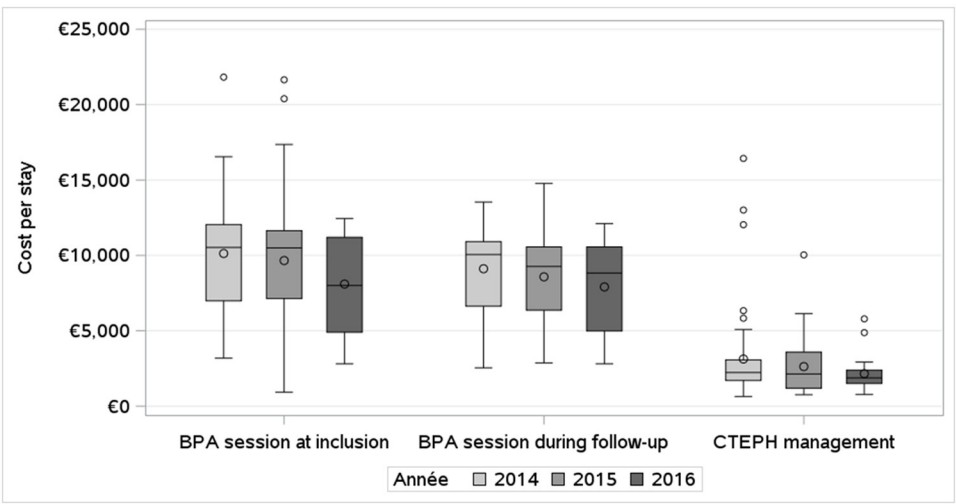

**Fig 2. Evolution of mean hospital cost per stay over year of inclusion.**

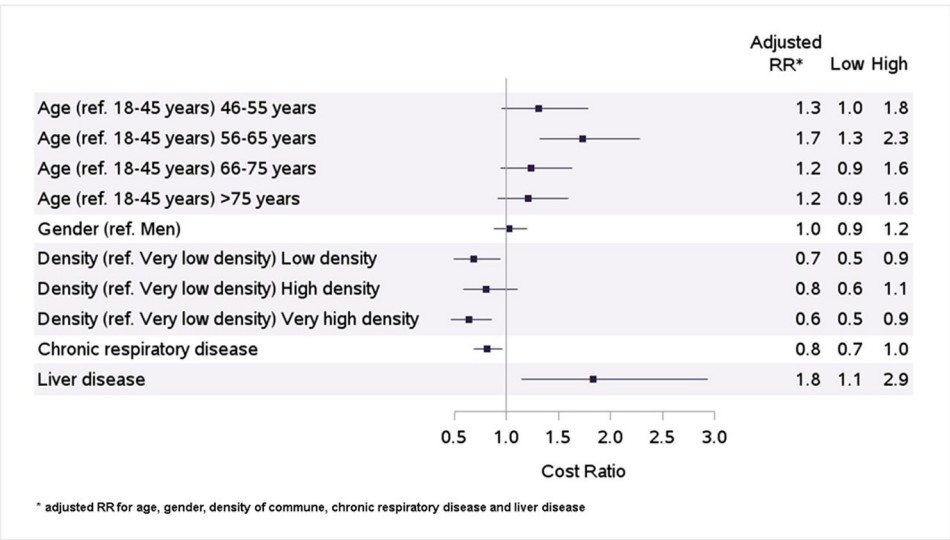

**Fig 3. Factors associated with hospital cost of BPA.**

## Discussion

To our knowledge, this is the first study assessing in real-life the hospital cost of BPA in patients with CTEPH.

Over a 6-month follow-up period, our study assessed a mean hospital cost for BPA of € 8,764 per stay and €21,245 per patient including both sessions and management. Costs were mainly driven by stays for BPA sessions. Complications did not represent a sizeable proportion of the cost. Costs tended to slightly decrease over the years with the reduction of length of stay, highlighting the learning curve of operators. Higher costs were observed for fragile patients (with an older age, with liver comorbidities) which may be explained by a natural heavier hospital management. Even if we excluded outlier patients, we cannot exclude some extra costs, not associated with BPA, that may still be present in our results. Higher costs were also reported for patients leaving in low-density area, even after exclusion of transportation costs. These higher costs may be explained by a potential delay in the diagnosis due to a later access to specialists for patients leaving far from cities and reference centres. These patients were then probably in more severe conditions at first BPA.

Some of these results could be discussed according to the existing literature. In a previous study, Brenot et al reported a series from the Paris Sud Centre, in which the medical records of 184 patients hospitalized for BPA at Paris Sud from February 2014 to July 2017 were analysed [8]. Our study included some patients who were reported in their series, as well as those from the Grenoble center. Despite expected similarities between the series by Brenot et al and the present study, some differences were observed. First, a history of PEA was identified for 3% of patients versus 8% in Brenot et al., suggesting that PEA may be underestimated in our study. The first BPA session was performed 1.1 years in median (IQR 0.3–2.92) after the first PH hospitalisation. The delay of 68 +/-51 months between PEA and BPA was observed which may be explained. One possible explanation is that we captured only the most recent PEA, occurring after 2006. In addition, an average of 2.8 BPA stays per patient were reported an average of 5.5 BPA sessions by patient in Brenot et al. According to experts' practice, 2 BPA sessions are usually performed by stay. Therefore around 5.6 BPA sessions by patient could be assumed in the present study. This result was in accordance with Brenot et al. publication. Finally, the

proportion of complications were quite low compared to Brenot et al. publication which may be explained by a management during BPA stays with session. Indeed, it seems we could not make a comparison between "stay for BPA complications" in our study and the actual incidence, because the majority of complications related to BPA were managed within the hospitalization during which the BPA is performed. This greatly relativizes both the number of stays for BPA management which are quite infrequent in our study and the value of the cost per patient for BPA management (which is ultimately very low).

Other results were in line with this publication: 121 patients were included between January 2014 and July 2016 (30 months) with the PMSI database in Paris Sud i.e around 4 patients/month versus 5 patients/month in Brenot et al study which included a more recent inclusion period. The mean age of patients was 64.3 years+/-14.7 versus 63 years+/-14 years in Brenot et al. and 53.1% were male versus 51% in Brenot et al. Furthermore, the mean length of stay for BPA of 6.7 days (incl. at inclusion and during follow-up), was consistent with what reported in a recent Japanese single center study of 125 patients who underwent BPA between November 2012 and September 2017 (6.6 days) [14]. We also found the same trend of shortening of hospital stay over years.

Our study presents however some limitations, mainly inherent to claims data sources. Some of them have no specific impact. As mentioned in the method section, CTEPH patients could not be identified directly because of the lack of specific ICD-10 codes for this rare disease. However, as BPA is a new therapeutic approach introduced in France in 2014 in only 2 centres of reference, a specific algorithm with the combination of ICD-10 codes for PH, specific procedure codes for BPA and hospital location of BPA session allowed to identify patients with a good accuracy. Moreover, the number of BPA sessions performed by BPA stays is not reported in PMSI. However, the cost of stay is valued based on DRG, therefore the number of procedures within a stay have no impact on stay valorisation. The study is focusing on the cost of BPA within its first years of implementation in France. However hospital practice and tariffs didn't change since this period. We also found in the study that complications stays were rare and should be even rarer over years. Therefore current hospital cost should not be different as those estimated in our study. The main limitation remained the lack of clinical data, as discussed above, potentially leading to lack of information regarding severity of the disease and underestimation of comorbidities. Since no information on clinical severity is available in the PMSI database, New York Hospital Association (NHYA) functional class cannot notably be assigned in our study as well as hemodynamic parameters and biomarkers. Comorbidities were estimated using ICD-10 codes reported in hospital stays occurring during the year prior to inclusion. Therefore, only recent and severe diseases (leading to hospitalization) were reported. Underreporting is therefore anticipated for these variables. In addition, costs could only be determined for in-hospital resource consumption, whereas community costs, such as primary care consultations, monitoring in community clinics and ambulatory prescription of PH-specific medication were not available.

## Conclusions

To conclude, our data suggested BPA had a mean hospital cost of € 8,764 per stay and €21,245 per patient within its first years of implementation in France. These results are especially interesting at a time when the position of BPA in the therapeutic algorithm is regularly discussed.

## Supporting information

**S1 File. ICD-10 codes of comorbidities.**
(DOCX)

**S2 File. Medical transports method.**
(DOCX)

## Acknowledgments

We thank Baptiste Jouaneton for his supervision of the overall operational conduct of the study and Charlène Tournier for the data extraction, data management and part of the statistical analysis. Both worked for HEVA at the time of the study conduct.

## Author Contributions

**Conceptualization:** Lionel Bensimon, Fanny Raguideau.

**Formal analysis:** Fanny Raguideau, Gwendoline Chaize.

**Methodology:** Vincent Cottin, Lionel Bensimon, Fanny Raguideau, Gwendoline Chaize, Jean Dallongeville, Hélène Bouvaist, Philippe Brenot.

**Project administration:** Lionel Bensimon.

**Software:** Gwendoline Chaize.

**Supervision:** Lionel Bensimon, Fanny Raguideau, Alexandre Vainchtock.

**Validation:** Vincent Cottin, Alexandre Vainchtock, Jean Dallongeville, Hélène Bouvaist, Philippe Brenot.

**Writing – original draft:** Fanny Raguideau.

**Writing – review & editing:** Vincent Cottin, Lionel Bensimon, Fanny Raguideau, Gwendoline Chaize, Antoinette Hakmé, Laurie Levy-Bachelot, Alexandre Vainchtock, Jean Dallongeville, Hélène Bouvaist, Philippe Brenot.

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
