## [Decision Letter · Decision Letter 0]

14 Jun 2021

PONE-D-21-11711

Hospital costs of Balloon Pulmonary Angioplasty (BPA) procedure and management for CTEPH patients: an observational study based on the French National hospital discharge database (PMSI)

PLOS ONE

Dear Dr. Raguideau,

Thank you for submitting your manuscript to PLOS ONE. After careful consideration, we feel that it has merit but does not fully meet PLOS ONE’s publication criteria as it currently stands. Therefore, we invite you to submit a revised version of the manuscript that addresses the points raised during the review process.

The reviewers were generally favorable, but did bring up some methodological concerns that will need to be addressed before this manuscript can be considered for publication. Please address each concern individually within the manuscript and detail the amendment made in a separate document. 

We look forward to receiving your revised manuscript.

Kind regards,

R. Jay Widmer

Academic Editor

PLOS ONE

Journal Requirements:

'This study was financed by MSD. LB, AH and LLB are employees of MSD and participated to the study design, study analysis, decision to publish and preparation of the manuscript. AV is one of the co-founders of the CRO HEVA; FR and GC are employees of HEVA, a company who received funding from the study sponsor for the conduct of this study. VC, JD, HB and PB are independent experts who received fees for participating in the scientific committee of the study.'

We note that one or more of the authors have an affiliation to the commercial funders of this research study:

MSD France and HEVA France.

a. Please provide an amended Funding Statement declaring these commercial affiliations, as well as a statement regarding the Role of Funders in your study. If the funding organization did not play a role in the study design, data collection and analysis, decision to publish, or preparation of the manuscript and only provided financial support in the form of authors' salaries and/or research materials, please review your statements relating to the author contributions, and ensure you have specifically and accurately indicated the role(s) that these authors had in your study. You can update author roles in the Author Contributions section of the online submission form.

b. Please also provide an updated Competing Interests Statement declaring these commercial affiliations along with any other relevant declarations relating to employment, consultancy, patents, products in development, or marketed products, etc.  

Within your Competing Interests Statement, please confirm that these commercial affiliations does not alter your adherence to all PLOS ONE policies on sharing data and materials by including the following statement: "This does not alter our adherence to  PLOS ONE policies on sharing data and materials.” (as detailed online in our guide for authors http://journals.plos.org/plosone/s/competing-interests). If this adherence statement is not accurate and  there are restrictions on sharing of data and/or materials, please state these. Please note that we cannot proceed with consideration of your article until this information has been declared.

Reviewers' comments:

Reviewer's Responses to Questions

**Comments to the Author**

1. Is the manuscript technically sound, and do the data support the conclusions?

Reviewer #1: Yes

Reviewer #2: No

2. Has the statistical analysis been performed appropriately and rigorously? 

Reviewer #1: Yes

Reviewer #2: Yes

3. Have the authors made all data underlying the findings in their manuscript fully available?

Reviewer #1: Yes

Reviewer #2: No

4. Is the manuscript presented in an intelligible fashion and written in standard English?

Reviewer #1: Yes

Reviewer #2: Yes

5. Review Comments to the Author

Reviewer #1: The authors performed a important study looking at the cost of BPA for CTEPH in France.

The analysis is well conducted with limitations that are adequately addressed in the discussion.

The manuscript is well written.

Minor suggestions:

1. In the discussion, line 217-218, the authors mentioned "additional with BPA session"; how was this determined in comparison to planned additional BPA sessions?

2. It would be helpful to have a description of the BPA planning in the methods section, ie how many sessions were planned with the number of session per admission, admission the night before the first BPA, etc. Also it would be very informative to know whether there was nay differences between the two centers performing BPA in France.

Reviewer #2: 1. Why are three MSD employees co-authors of this study? I would rather have a CTEPH surgeon included, and some other members of the local CTEPH teams. BPA is a team decision and that should be reflected in authorships. MSD employees have a conflict of interest that cannot be overcome.

2. In the introduction the market approval of subcutaneous Treprostinil for the treatment of not operated CTEPH, and persistent recurrent PH after pulmonary endarterectomy should be mentioned. The guideline recommendation is of 2015 and today outdated.

3. The period for which the cost analysis was performed includes the French BPA learning curve and may not be representative of today‘s cost. Authors should calculate cost for a cohort that is more contemporary, for example 2018-2019.

4. What if diagnostic procedures were done in the same hospital stay as the first BPA SESSION? Were these stays counted as BPAs? Particularly in patients from distance diagnosis and treatments are oftentimes performed within the same stay.

5. …follow-up stays had a mean length of 6.0 days…Authors need to provide a precise definition of follow-up stays. Is this corresponding to stays after the first BPA session? Why 6 days, and later 2?

6. Were cost for Drugs included, or paid by healthcare plans outside the hospital?

7. Was riociguat given?

8. How often could riociguat or prostacyclins be stopped because of successful BPA?

9. Authors should calculate annual cost of vasodilatior therapy given to patients prior to and after BPA. This should serve as a comparator because these treatments would be the placeholder for BPA if that were not paid for.

10. Transportation cost should be covered by the patients.

11. A mean hospital cost for BPA of € 8,764 per stay and two BPA procedures is very expensive in my eyes. For BPA one needs 200mL contrast, a guiding sheath, a guiding catheter, a wire and a few balloons at a cost of € 30 each. Was peripheral or coronary material used? Were any CTOs done?

12. What was procedural cost? Was it different between Paris and Grenoble?

13. What was included in the SOPs for lung injury? Were chest CT scans mandatory?

6. PLOS authors have the option to publish the peer review history of their article (what does this mean?). If published, this will include your full peer review and any attached files.

Reviewer #1: No

Reviewer #2: No

---

## [Author Response · Author response to Decision Letter 0]

18 Aug 2021

Editor comments

1. Please ensure that your manuscript meets PLOS ONE's style requirements, including those for file naming. The PLOSONE style templates can be found at

Answer to response 1 :

We verified that the manuscript meets PLOS ONES’s style requirements

Upon re-submitting your revised manuscript, please upload your study’s minimal underlying data set as either Supporting Information files or to a stable, public repository and include the relevant URLs, DOIs, or accession numbers within your revised cover letter. For a list of acceptable repositories, please see http://journals.plos.org/plosone/s/dataavailability# loc-recommended-repositories. Any potentially identifying patient information must be fully anonymized.

Important: If there are ethical or legal restrictions to sharing your data publicly, please explain these restrictions in detail.

Please see our guidelines for more information on what we consider unacceptable restrictions to publicly sharing data:http://journals.plos.org/plosone/s/data-availability#loc-unacceptable-data-access-restrictions. Note that it is not acceptable for the authors to be the sole named individuals responsible for ensuring data access. We will update your Data Availability statement to reflect the information you provide in your cover letter.

Answer to response 2 :

Due to the sensitive nature of the data that support the findings, access to them is restricted and can only be granted by the Ethics and scientific committee for health research, studies, and evaluations (CESREES, Comité Ethique et Scientifique pour les Recherches, les Etudes et les Evaluations dans le domaine de la Santé) and/or the French data protection authority (Comité National de l’Informatique et des Libertés, CNIL), and so are not readily available. The data are part of the National health data system (SNDS, Système national des Données de Santé) a database maintained by the HDH (Health Data Hub). The HDH can be contacted at https://www.health-data-hub.fr/contact. 

'This study was financed by MSD. LB, AH and LLB are employees of MSD and participated to the study design, review of the protocol, publication planning and preparation of the manuscript. AV is one of the co-founders of the CRO HEVA; FR and GC are employees of HEVA, a company who received funding from the study sponsor for the conduct of this study. VC, JD, HB and PB are independent experts who received fees for participating in the scientific committee of the study.'

We note that one or more of the authors have an affiliation to the commercial funders of this research study:

MSD France and HEVA France.

a. Please provide an amended Funding Statement declaring these commercial affiliations, as well as a statement regarding the Role of Funders in your study. If the funding organization did not play a role in the study design, data collection and analysis, decision to publish, or preparation of the manuscript and only provided financial support in the form of authors' salaries and/or research materials, please review your statements relating to the author contributions, and ensure you have specifically and accurately indicated the role(s) that these authors had in your study. You can update author roles in the Author Contributions section of the online submission form.

Please also include the following statement within your amended Funding Statement. “The funder provided support in the form of salaries for authors [insert relevant initials], but did not have any additional role in the study design, data collection and analysis, decision to publish, or preparation of the manuscript. The specific roles of these authors are articulated in the ‘author contributions’ section.”

Answer to response 3.a. 

The funder allowed to perform the study. The funder provided support in the form of salaries for authors [LB,AH and LLB], but did not have any additional role in the study design, data collection and analysis, decision to publish, or preparation of the manuscript. The specific roles of these authors are articulated in the ‘author contributions’ section.”

b. Please also provide an updated Competing Interests Statement declaring these commercial affiliations along with any other relevant declarations relating to employment, consultancy, patents, products in development, or marketed products, etc.

Within your Competing Interests Statement, please confirm that these commercial affiliations does not alter your adherence to all PLOS ONE policies on sharing data and materials by including the following statement: "This does not alter our adherence to PLOS ONE policies on sharing data and materials.” (as detailed online in our guide for authors

http://journals.plos.org/plosone/s/competing-interests). If this adherence statement is not accurate and there are restrictions on sharing of data and/or materials, please state these. Please note that we cannot proceed with consideration of your article until this information has been declared.

Answer to response 3.b. 

We updated Competing Interests Statement declaring that these commercial affiliations does not alter our adherence to all PLOS ONE policies on sharing data and materials (detail below).

Please know it is PLOS ONE policy for corresponding authors to declare, on behalf of all authors, all potential competing interests for the purposes of transparency. PLOS defines a competing interest as anything that interferes with, or could reasonably be perceived as interfering with, the full and objective presentation, peer review, editorial decision-making, or publication of research or non-research articles submitted to one of the journals. Competing interests can be financial or non-financial, professional, or personal. Competing interests can arise in relationship to an organization or another person. Please follow this link to our website for more details on competing interests:

http://journals.plos.org/plosone/s/competing-interests.

Answer to response 3.c. 

We provided an updated Funding Statement and Competing Interests Statement in the cover letter (and detail below).

Competing interests

I have read the journal's policy and the authors of this manuscript have the following competing interests:

Experts: 

VC, JD, HB and PB are independent experts who received fees for participating in the scientific committee of the study. Outside the submitted work, VC reports personal fees and non-financial support from Actelion, grants, personal fees and non-financial support from Boehringer Ingelheim, personal fees from Bayer / MSD, personal fees from Novartis, personal fees and non-financial support from Roche / Promedior, personal fees from Sanofi, personal fees from Celgene / BMS, personal fees from Galapagos, personal fees from Galecto, personal fees from Shionogi, personal fees from Astra Zeneca, personal fees from Fibrogen, personal fees from RedX, personal fees from PureTech. Outside the submitted work, HB has received personal fees and/or non-financial support from MSD, Actelion, GSK. Outside the submitted work, PB and JD have no competing interest in relation with this topic to declare.

MSD employees:

LB, AH and LLB are employees of MSD, the company which financed the study. 

HEVA employees:

AV is one of the co-founders of the CRO HEVA; FR and GC are employees of HEVA, a company who received funding from the study sponsor for the conduct of this study and for the analysis of the data. 

These commercial affiliations does not alter adherence to all PLOS ONE policies on sharing data and materials.

Financial statement

This study was financed by MSD. The funder provided support in the form of salaries for authors [LB,AH and LLB], but did not have any additional role in the study design, data collection and analysis, decision to publish, or preparation of the manuscript. LB, AH and LLB are employees of MSD and participated to the study design, study analysis, decision to publish and preparation of the manuscript. AV is one of the co-founders of the CRO HEVA; FR and GC are employees of HEVA, a company who received funding from the study sponsor for the conduct of this study. VC, JD, HB and PB are independent experts who received fees for participating in the scientific committee of the study.

Reviewers' comment

Reviewer #1: The authors performed a important study looking at the cost of BPA for CTEPH in France.

The analysis is well conducted with limitations that are adequately addressed in the discussion.

The manuscript is well written.

Minor suggestions:

1. In the discussion, line 217-218, the authors mentioned "additional with BPA session"; how was this determined in comparison to planned additional BPA sessions?

Answer to response 1 :

Thank you for the positive assessment of the study and manuscript. CTEPH patients were included at time of first BPA session (index date). However BPA sessions are iterative and there is possibility of revision at distance of first intervention. Stays with a BPA session occurring after the index date (defined as the first stay with a BPA session) and during the 6 month - follow up period, were defined as additional BPA session in the manuscript. We used the same algorithm to identify first BPA session and additional BPA session in the data set. 

The line 217-218 was not modified but we clarified the term “additional” before in the text in Material and methods section (line 144-146): 

“Stays with BPA sessions (inclusion and additional stays) were identified as stays with both a diagnosis code (as PD or RD or SAD) for PH and a BPA procedure performed in Paris Sud and/or Grenoble, as per inclusion criteria.“

2. It would be helpful to have a description of the BPA planning in the methods section, ie how many sessions were planned with the number of session per admission, admission the night before the first BPA, etc. Also it would be very informative to know whether there was any differences between the two centers performing BPA in France.

Answer to response 2 :

In France, the patient is admitted the night before the intervention. Two sessions of BPA are planned separated from 48 hours. Patient stays around 6 days at hospital. This protocol of BPA is the same in all French centers, and notably in Paris Sud University and Grenoble University hospital. 

We described the protocol in Material and methods section (line 102-104):

“ BPA protocol is the same in both centers. Patient spent around 6 days at the hospital with an overnight admission for two BPA sessions repeated within 48 hours.“ 

Reviewer #2: 

1. Why are three MSD employees co-authors of this study? I would rather have a CTEPH surgeon included, and some other members of the local CTEPH teams. BPA is a team decision and that should be reflected in authorships. MSD employees have a conflict of interest that cannot be overcome.

Answer to response 1:

Regarding MSD employees co-authors, we included all persons that contributed to this work in accordance with ICJME guidelines. Their role in the study was the following: Laurie Levy-Bachelot wad the project director, Lionel Bensimon was the project leader and Antoinette Hakmé was the medical leader. Your suggestion to involve a surgeon is excellent. Unfortunately, as the study has already been performed, it is no longer possible to add authors who have not yet participated.

2. In the introduction the market approval of subcutaneous Treprostinil for the treatment of not operated CTEPH, and persistent recurrent PH after pulmonary endarterectomy should be mentioned. The guideline recommendation is of 2015 and today outdated.

Answer to response 2:

Thank you for your comment. Please not, that Treprostinil (as mentioned in the ERS statement 2021 ) was not EU approved for CTEPH by the time of the study and is still not reimbursed in France for this indication. 

However, since the goal of the study is to assess the "Hospital costs of Balloon Pulmonary Angioplasty (BPA) procedure and management for CTEPH patients" and not the particular effect of one or other drug we agree with the reviewer describing drug therapies in the introduction of the manuscript may be confusing as it was not included in this work. Therefore in order to avoid any confusion we decided to remove all parts mentioning drug therapies. 

3. The period for which the cost analysis was performed includes the French BPA learning curve and may not be representative of today‘s cost. Authors should calculate cost for a cohort that is more contemporary, for example 2018-2019.

Answer to response 3:

You are completely right. As raised in the conclusion, our study is focusing on the cost of BPA within its first years of implementation in France. We agree that it should be interesting to update the study with additional years to see the evolution of costs in recent years. Unfortunately it was not planned at the beginning of the project and then we do not have these data in our dataset and it takes time to perform and interpret the results anyway.

However in France, hospital cost estimation is based on a fixed national tariffication from the Social Security that apply to each hospitalization anywhere in France, and based on the Diagnosis Related Group code attributed in the PMSI database. These standard tariffs include medical and related procedures, nursing care, treatments (except specific expensive drugs), food and accommodation, and investment costs for hospitalised patients. The tariff didn’t change since the initiation of the studied period. Furthermore, BPA hospital practice didn’t change either. Finally the complications were and remained rare over years. Therefore it is unlikely that the current hospital cost differs substantially from those estimated in our study. 

But as it’s an important point we added this limitation in the Discussion as follows : 

“The study is focusing on the cost of BPA within its first years of implementation in France. However hospital practice and tariffs didn’t change since this period. We also found in the study that complications stays were rare and should be even rarer over years. Therefore current hospital cost should not be different as those estimated in our study. “

4. What if diagnostic procedures were done in the same hospital stay as the first BPA SESSION? Were these stays counted as BPAs? Particularly in patients from distance diagnosis and treatments are oftentimes performed within the same stay.

Answer to response 4:

If diagnostic procedures are performed within the same hospital stay that the first BPA session the cost is including in the cost of the stay. 

5. …follow-up stays had a mean length of 6.0 days…Authors need to provide a precise definition of follow-up stays. Is this corresponding to stays after the first BPA session? Why 6 days, and later 2?

Answer to response 5:

Apologies for the lack of clarity. Follow up stays are stays that occurred during the follow up period, after the first BPA session. The mean length of stay for BPA session follow up stays was 6.0 days. However, the mean length of stay for BPA management follow up stays was 2.0 days. 

We modified the manuscript in order to precise this point (line 235-239):

“The first stays with BPA sessions (inclusion stays) had a mean length of 7.8 days (SD=4.5) whereas follow-up stays with BPA sessions had a mean length of 6.0 days (SD=2.3). Moreover, length of BPA stays decreased over the years with a mean of 8.5 days (SD= 4.6) in 2014, 7.9 (SD= 3.8) in 2015 and 6.9 (SD= 5.4) in 2016. Follow-up stays for BPA management and CTEPH management were shorter with a mean length of 2.0 days (SD=1.4) and 3.7 days (SD=3.7), respectively.”

6. Were cost for Drugs included, or paid by healthcare plans outside the hospital?

Answer to response 6:

All drugs were paid by the healthcare plan and are included in the fixed national tariffication from the Social Security for a hospital stay. Of note, the study was performed using the French hospital discharge database (PMSI). Therefore only healthcare occurring at hospitals are available in this database. Regarding drugs, most drugs cannot be specifically identified since they are integrated into the Diagnostic Related Group (DRG) cost. However, delivery of certain expensive drugs can be identified individually. In this study we only focused on hospital stay costs. This point was raised in the discussion but we propose to be more specific in order to highlight the drugs mentioned in your next comments. 

The sentence will be modified as follows :

« In addition, costs could only be determined for in-hospital resource consumption, whereas community costs, such as primary care consultations, monitoring in community clinics and ambulatory prescription of PH-specific medication were not available ». 

7. Was riociguat given?

Answer to response 7:

Riociguat is not available at hospital and was therefore not studied. We propose to modify the sentence in the discussion to precise this point.

The sentence will be modified as follows :

« In addition, costs could only be determined for in-hospital resource consumption, whereas community costs, such as primary care consultations, monitoring in community clinics and ambulatory prescription of PH-specific medication were not available ». 

8. How often could riociguat or prostacyclins be stopped because of successful BPA?

Answer to response 8:

These treatment are not available at hospital. We can’t perform the requested analysis that is beyond the aim of the present study.

We have modified the sentence in the discussion as follows.

« In addition, costs could only be determined for in-hospital resource consumption, whereas community costs, such as primary care consultations, monitoring in community clinics and ambulatory prescription of PH-specific medication were not available ». 

9. Authors should calculate annual cost of vasodilatior therapy given to patients prior to and after BPA. This should serve as a comparator because these treatments would be the placeholder for BPA if that were not paid for.

Answer to response 9:

Vasodilatator therapy are not available at hospital. We can’t perform the requested analysis. Anyway, the hospital cost is not impacted by treatment cost in the current analysis as a fixed national tariffication from the Social Security is used.

We have modified the sentence in the discussion as follows.

« In addition, costs could only be determined for in-hospital resource consumption, whereas community costs, such as primary care consultations, monitoring in community clinics and ambulatory prescription of PH-specific medication were not available ». 

10. Transportation cost should be covered by the patients.

Answer to response 10:

In France, medical transports to hospital are reimbursed by National Health Insurance for patients with severe conditions such as CTEPH. In this study, the costs were estimated from the National Health Insurance perspective, that is the reason why medical transport costs are included. However, we provide in the manuscript the detail of the cost with and without transport. Please kindly refer to the manuscript , Results/page 13. 

11. A mean hospital cost for BPA of € 8,764 per stay and two BPA procedures is very expensive in my eyes. For BPA one needs 200mL contrast, a guiding sheath, a guiding catheter, a wire and a few balloons at a cost of € 30 each. Was peripheral or coronary material used? Were any CTOs done?

Answer to response 11:

The hospital cost for BPA includes all costs provided during hospitalization for BPA i.e. medical and related procedures, nursing care, treatments (except specific expensive drugs), food and accommodation, and investment costs for hospitalized patients., not only the material used to perform BPA. In agreement with the reviewer's feeling the costs were driven by hospital stays, not by material use.

12. What was procedural cost? Was it different between Paris and Grenoble?

Answer to response 12:

The procedural cost is defined by the National Health Insurance. It’s the same in all French hospital centers, and notably in Paris Sud University and Grenoble University hospital.

13. What was included in the SOPs for lung injury? Were chest CT scans mandatory?

Answer to response 13:

Chest CT scans are not carried out systematically. Chest radiography can be performed prior to the procedure. However, carrying out these examinations has no significant impact on the hospital cost as to a very larger extent length of stay or hospitalization in an intensive care unit are the major drivers of hospital cost in the present study.

---

## [Decision Letter · Decision Letter 1]

8 Sep 2021

PONE-D-21-11711R1

Hospital costs of Balloon Pulmonary Angioplasty (BPA) procedure and management for CTEPH patients: an observational study based on the French National hospital discharge database (PMSI)

PLOS ONE

Dear Dr. Raguideau,

Thank you for submitting your manuscript to PLOS ONE. After careful consideration, we feel that it has merit but does not fully meet PLOS ONE’s publication criteria as it currently stands. Therefore, we invite you to submit a revised version of the manuscript that addresses the points raised during the review process.

Please address the final few comments from Reviewer #2 individually to satisfy requirements for acceptance. We thank you for your diligence in creating your first revision paying careful attention to each comment. 

We look forward to receiving your revised manuscript.

Kind regards,

R. Jay Widmer

Academic Editor

PLOS ONE

Journal Requirements:

Additional Editor Comments (if provided):

The reviewers were mostly favorable toward the revised submission. Please address the final few items from reviewer#2 individually.

Reviewers' comments:

Reviewer's Responses to Questions

**Comments to the Author**

1. If the authors have adequately addressed your comments raised in a previous round of review and you feel that this manuscript is now acceptable for publication, you may indicate that here to bypass the “Comments to the Author” section, enter your conflict of interest statement in the “Confidential to Editor” section, and submit your "Accept" recommendation.

Reviewer #1: All comments have been addressed

Reviewer #2: (No Response)

2. Is the manuscript technically sound, and do the data support the conclusions?

Reviewer #1: Yes

Reviewer #2: Yes

3. Has the statistical analysis been performed appropriately and rigorously? 

Reviewer #1: Yes

Reviewer #2: Yes

4. Have the authors made all data underlying the findings in their manuscript fully available?

Reviewer #1: Yes

Reviewer #2: No

5. Is the manuscript presented in an intelligible fashion and written in standard English?

Reviewer #1: Yes

Reviewer #2: Yes

6. Review Comments to the Author

Reviewer #1: The authors have adequately addressed my concerns.

This paper will be an important contribution to the field.

Reviewer #2: This is a interesting paper that deserves to be published. However,

1. I do not agree with the response to qu 2. One cannot negate that drugs are part of CTEPH management. Therefore, drugs should be mentioned in the introduction, including their availability in France. Authors will need to clarify that drugs costs are not subject of the present analysis as they were paid out-of-hospital.

2. I realized that the important message within authors‘ responses that …intensive care unit are the major drivers of hospital cost in the present study….cannot easily be found in the manuscript. Please, add those cost for intensive care stays to Table 2.

7. PLOS authors have the option to publish the peer review history of their article (what does this mean?). If published, this will include your full peer review and any attached files.

Reviewer #1: No

Reviewer #2: No

---

## [Author Response · Author response to Decision Letter 1]

21 Oct 2021

Reviewer #1: The authors have adequately addressed my concerns.

This paper will be an important contribution to the field.

Reviewer #2: 

This is an interesting paper that deserves to be published. However,

1. I do not agree with the response to question 2. One cannot negate that drugs are part of CTEPH management. Therefore, drugs should be mentioned in the introduction, including their availability in France. Authors will need to clarify that drugs costs are not subject of the present analysis as they were paid out-of-hospital.

Response to answer 1 to reviewer 2:

We added the drug authorized for CTEPH management in the introduction part (page 3, lines 64-66) as above :

“Riociguat, an oral guanylate cyclase stimulator, and treprostinil, a subcutaneous prostacyclin analogue, are approved (in 2014 and 2020 respectively) for patients with inoperable CTEPH or persistent/recurrent PH after PEA; other PH medications are also off-label used(6)”

We also precised in the methodology part of the manuscript that out of hospital drugs are not analysed in the present study (page 7, lines 158-159):

“Out of hospital drugs were not included in the present analysis as they are not recorded in the database.”

6. Delcroix M, Torbicki A, Gopalan D, Sitbon O, Klok FA, Lang I, et al. ERS statement on chronic thromboembolic pulmonary hypertension. Eur Respir J. 2021 Jun;57(6):2002828. 

2. I realized that the important message within authors‘ responses that …intensive care unit are the major drivers of hospital cost in the present study….cannot easily be found in the manuscript. Please, add those cost for intensive care stays to Table 2.

Response to answer 2 to reviewer 2:

We agree that this information is interesting to add. Therefore, we added the following relevant results regarding ICU in table 2 : number of stays with ICU transit ; mean length of stay in ICU, part of ICU cost among total cost, mean ICU cost/stay and mean ICU cost/patient. We also modified the text in consequence to describe these results.

---

## [Decision Letter · Decision Letter 2]

11 Nov 2021

Hospital costs of Balloon Pulmonary Angioplasty (BPA) procedure and management for CTEPH patients: an observational study based on the French National hospital discharge database (PMSI)

PONE-D-21-11711R2

Dear Dr. Raguideau,

We’re pleased to inform you that your manuscript has been judged scientifically suitable for publication and will be formally accepted for publication once it meets all outstanding technical requirements.

Kind regards,

R. Jay Widmer

Academic Editor

PLOS ONE

Reviewers' comments:

Reviewer's Responses to Questions

**Comments to the Author**

1. If the authors have adequately addressed your comments raised in a previous round of review and you feel that this manuscript is now acceptable for publication, you may indicate that here to bypass the “Comments to the Author” section, enter your conflict of interest statement in the “Confidential to Editor” section, and submit your "Accept" recommendation.

Reviewer #2: All comments have been addressed

2. Is the manuscript technically sound, and do the data support the conclusions?

Reviewer #2: Yes

3. Has the statistical analysis been performed appropriately and rigorously? 

Reviewer #2: Yes

4. Have the authors made all data underlying the findings in their manuscript fully available?

Reviewer #2: Yes

5. Is the manuscript presented in an intelligible fashion and written in standard English?

Reviewer #2: Yes

6. Review Comments to the Author

Reviewer #2: no further comments no further comments no further comments no further comments no further comments no further comments

7. PLOS authors have the option to publish the peer review history of their article (what does this mean?). If published, this will include your full peer review and any attached files.

Reviewer #2: No

---

## [Editor Report · Acceptance letter]

16 Nov 2021

PONE-D-21-11711R2 

Hospital costs of Balloon Pulmonary Angioplasty (BPA) procedure and management for CTEPH patients: an observational study based on the French National hospital discharge database (PMSI) 

Dear Dr. Raguideau:

I'm pleased to inform you that your manuscript has been deemed suitable for publication in PLOS ONE. Congratulations! Your manuscript is now with our production department. 

Kind regards, 

on behalf of

Dr. R. Jay Widmer 

Academic Editor

PLOS ONE